# A Baseline Method for Removing Invisible Image Watermarks using Deep Image Prior

**Hengyue Liang**                                    *liang656@umn.edu*
*Electrical and Computer Engineering*
*University of Minnesota, Twin Cities*

**Taihui Li**                                    *lixx5027@umn.edu*
*Computer Science and Engineering*
*University of Minnesota, Twin Cities*

**Ju Sun**                                    *jusun@umn.edu*
*Computer Science and Engineering*
*University of Minnesota, Twin Cities*

**Reviewed on OpenReview:** *https://openreview.net/forum?id=g85Vxlrq0O*

## Abstract

Image watermarks have been considered a promising technique to help detect AI-generated content, which can be used to protect copyright or prevent fake image abuse. In this work, we present a black-box method for removing *invisible* image watermarks, without the need of any dataset of watermarked images or any knowledge about the watermark system. Our approach is simple to implement: given a *single* watermarked image, we regress it by deep image prior (DIP). We show that from the intermediate steps of DIP one can reliably find an evasion image that can remove invisible watermarks while preserving high image quality. Due to its unique working mechanism and practical effectiveness, we advocate including DIP as a baseline invasion method for benchmarking the robustness of watermarking systems. Finally, by showing the limited ability of DIP and other existing black-box methods in evading training-based *visible* watermarks, we discuss the positive implications on the practical use of training-based *visible* watermarks to prevent misinformation abuse. Our code is publicly available at: `https://github.com/sun-umn/DIP_Watermark_Evasion_TMLR`.

## 1 Introduction

In this prosperous era of generative AI, the traceability of AI-generated content (e.g., language, images, and videos) to its source has been frequently mentioned as a promising solution to promote the responsible use of generative AI (Fan et al., 2023), e.g., to protect copyright or to curb misinformation. In particular, the traceability of AI-generated images has become increasingly urgent, as many AI products, such as DALL-E (Ramesh et al., 2022) and Stable Diffusion (Rombach et al., 2022), can create highly photorealistic and artistic images that are hard to distinguish from natural photos or human drawings. Unsurprisingly, some AI-generated images have caused false beliefs on social media (Wendling). Thus, major tech companies such as Google, OpenAI (Bartz & Hu, 2023) have recently opted to incorporate watermarks into their image generation products to improve traceability and promote responsible use.

**Manually designed vs. training-based watermarks** Watermarking methods can be generally divided into two categories: **(i)** *non-blind* methods (Cox et al., 1997; Hsieh et al., 2001; Pereira & Pun, 1999) and **(ii)** *blind* methods (Bi et al., 2007), which are divided by whether access to original clean images is required to correctly decode the watermarked images (Zhao et al., 2024a). In what follows, we focus only on blind methods (and refer to them as *watermarks*), as they do not require access to clean images and fit better in

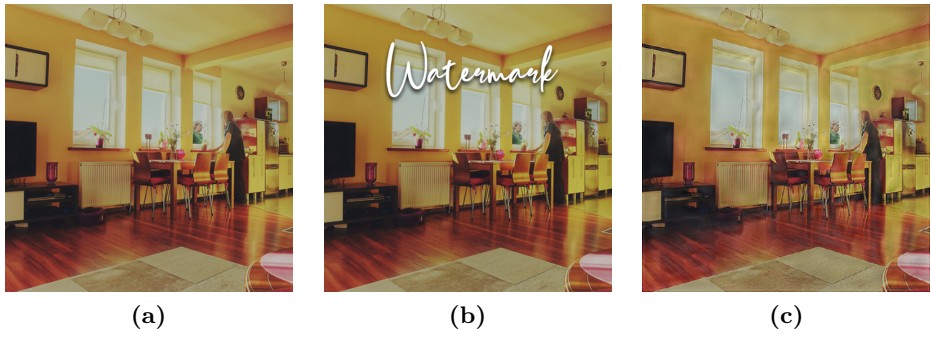

**(a)**          **(b)**          **(c)**

Figure 1: Example of (a) a clean image, (b) an image with an overlaid logo watermark and (c) an image with steganography watermark.

large-scale application scenarios, such as tracing AI-generated content. Watermarks are typically embedded in images in terms of an overlaid logo or steganography[1] (Morkel et al., 2005); see Fig. 1 for an example. The embedded watermarks then are typically detected by human eyes or by algorithmic decoders (Voyatzis & Pitas, 1999; Zhu et al., 2018). In early research, various manually designed watermarks were proposed and applied to copyright protection, e.g., visible watermarks (detectable by human eyes) such as an overlaid layer (Kankanhalli et al., 1999) or a color code signature (such as the example in Appendix A); invisible watermarks (detectable by algorithmic decoders) such as Tirkel et al. (1993); Pereira & Pun (2000); Navas et al. (2008). However, these manually designed watermarks are challenged by robustness concerns, i.e., imperceptible corruption, digital editing, or deliberate attacks to the watermarked image can make them undetectable (Mishra, 2022; Zhao et al., 2024b); see also Fig. 2. To address these challenges, recent work has shifted to training-based watermarking systems using deep neural networks (DNNs). By incorporating ideas from data augmentation (Mumuni & Mumuni, 2022) and adversarial training (Goodfellow et al., 2014b), the robustness of training-based watermarks against common digital corruptions consistently beats that of manually designed ones (Zhu et al., 2018; Zhang et al., 2019b; Tancik et al., 2020; Jia et al., 2021).

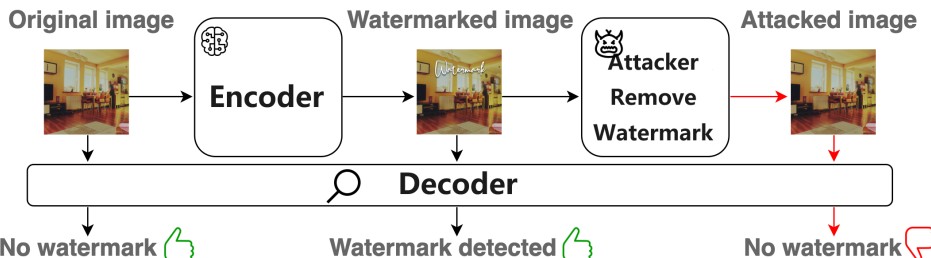

Figure 2: Illustration of the watermarking system and the robustness concern—watermarks may be removed with little loss of image quality.

**Evaluating the robustness of watermark systems** Robustness evaluation of watermark systems often operates under either the white-box or the black-box evasion (i.e., attack) setting (Zhao et al., 2024a; An et al., 2024). In white-box evasion, the decoder and the groundtruth watermark are accessible to a dedicated evader (Jiang et al., 2023), which easily leads to a high threat. In contrast, the threat can be greatly reduced by maintaining secrecy of the system (An et al., 2024), i.e., under the *black-box* model, where the decoder and the groundtruth watermark remain unknown to the evader. Nowadays, it is common for companies not to open-source their generative products, e.g., DALLE-3 from OpenAI, Midjourney, Imagen from Google. Thus, being robust against potential black-box evasions is of higher priority in practical scenarios.

The main focus of this paper is to explore *black-box* evasion techniques that can generate evasion images with the best possible quality. We stress the image quality alongside the evasion success, because for a dedicated evader, the higher the quality of the evasion images they can produce, the more benefit they can potentially

---

[1]Steganography: detectable messages embedded in an image but are invisible to human eyes.

gain, e.g., breaking the copyright protection without compromising the image quality or misleading people to use highly naturally-looking fake images. Recently, researchers are putting effort into standardizing the robustness benchmark of image watermarks (An et al., 2024). Although the selected black-box evasion methods are effective in bypassing watermark detection, their evasion images can still introduce visible defects; see Section 4 and Fig. 8. Thus, the search for new methods to provide stronger stress tests of watermarking systems is still a pressing problem.

**Our contributions**   In this paper, we propose a new black-box watermark evasion method using the Deep Image Prior (DIP) (Ulyanov et al., 2018)—an untrained DNN-based prior that proves powerful in solving single-image blind denoising and numerous single-instance inverse problems (Qayyum et al., 2023; Tirer et al., 2023; Zhuang, 2023). Our main contributions include: **(i)** We show that DIP-based blind denoising can be used to generate high-quality evasion images effective against many existing invisible watermarks, both training-based and manually designed; **(ii)** We elucidate the principle behind DIP's evasion performance—its faster rate in picking up low frequencies than high ones empowers its image-agnostic watermark purification ability. Due to **(i)** and **(ii)**, we advocate including DIP evasion as an integral component in the robustness evaluation of watermarking systems (Saberi et al., 2023; An et al., 2024); **(iii)** Based on our analysis, we further recommend that to counteract black-box watermark evasions, a reliable watermark scheme should focus on modifying low-frequency components and have a reasonable magnitude.

## 2   Background and related work

**(Blind) image steganography**   refers to the technique of hiding secret but retrievable messages in an image with minimal change to the image (Zhu et al., 2018). Given an arbitrary natural image $I \in \mathcal{I}$, where $\mathcal{I}$ denotes the set of natural images, and an arbitrary $n$-bit message $\boldsymbol{w} \in \{0, 1\}^n$, an image steganography system typically consists of an encoder $E$—which takes any image $I$ and any message $\boldsymbol{w}$ and produces an encoded image, a decoder $D$—which takes any image and produces an informative message, and its system goal: ($\circ$ means function composition)

$$
\begin{aligned}
&(D \circ E)(I, \boldsymbol{w}) = \boldsymbol{w}, && \forall I \in \mathcal{I}, \forall \boldsymbol{w} \in \{0, 1\}^n, && \text{(correctly encode and decode } \boldsymbol{w}) && \text{(1a)} \\
&D(I) = \emptyset, && \forall I \in \mathcal{I}, && \text{(no useful message decoded from a clean image)} && \text{(1b)} \\
&E(I, \boldsymbol{w}) \approx I, && \forall I \in \mathcal{I}, \forall \boldsymbol{w} \in \{0, 1\}^n. && \text{(minimal encoding distortion to the image)} && \text{(1c)}
\end{aligned}
$$

Existing steganography methods differ by whether the encoder and decoder are manually designed or learned from data. Manually designed encoder-decoder pairs rely on ideas such as manipulating the least significant bit (LSB) (Tirkel et al., 1993), template matching in the Fourier domain (Pereira & Pun, 2000), discrete wavelet transform (DWT), discrete cosine transform (DCT), and singular value decomposition (SVD) (Bi et al., 2007; Pereira & Pun, 2000; Navas et al., 2008). In contrast, training-based methods often learn DNN-based encoder-decoder pairs from data, based on variants of a model formulation derived from the goal stated in Eq. (1):

$$
\begin{aligned}
&\min_{\boldsymbol{\phi}, \boldsymbol{\theta}} \mathbb{E}_{\boldsymbol{w}, I} \ell_m[\boldsymbol{w}, (D_{\boldsymbol{\theta}} \circ E_{\boldsymbol{\phi}})(I, \boldsymbol{w})] && \text{(to ensure Eq. (1a))} \\
&\text{s.t. } \ell_q(I, E_{\boldsymbol{\phi}}(I, \boldsymbol{w})) \leq \delta, \quad \forall I \in \mathcal{I}, \ \forall \boldsymbol{w} \in \{0, 1\}^n, && \text{(to ensure Eq. (1c))}
\end{aligned}
\tag{2}
$$

where $\boldsymbol{\phi}$ and $\boldsymbol{\theta}$ are learnable weights of the DNNs in $E$ and $D$, respectively; $\ell_m$ and $\ell_q$ are two losses measuring the error of **m**essage recovery and the **q**uality distortion to the image, respectively; and $\delta$ is the maximally allowed perturbation to the image caused by watermarking embedding. Representative training-based methods include HIDDEN and its variants (Zhu et al., 2018; Wen & Aydore, 2019; Luo et al., 2020), SteganoGAN (Zhang et al., 2019a), Stable Signature (Fernandez et al., 2023), rivaGAN (Zhang et al., 2019b), StegaStamp (Tancik et al., 2020), Mbrs (Jia et al., 2021) and TrustMark (Bui et al., 2023a). These methods typically also incorporate regularization terms to encourage the distribution of the encoded images to be close to that of the original images based on generative adversarial networks (GAN) (Goodfellow et al., 2014a). SSL (Fernandez et al., 2022) and RoSteALS (Bui et al., 2023b) are similar in spirit but perform learning in different spaces. In theory, solving Eq. (2) with a reasonably small $\delta$ can always produce distortion patterns that are *invisible*

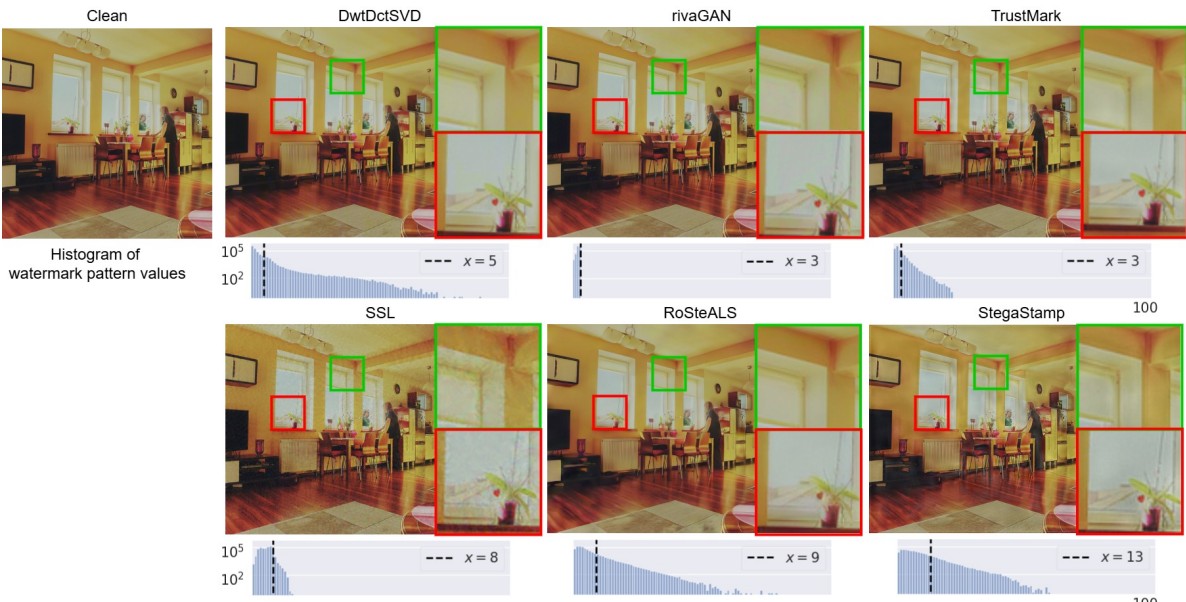

Figure 3: Visualization of a clean image (from COCO) and its steganography versions by different methods. Below each steganography image is the histogram of the pixel magnitude in the corresponding distortion (i.e., difference between the steganography image and the clean image): $y$-axis is in log scale and the vertical dashed line marks the 90% quantile. While DwtDctSVD, rivaGAN and TrustMark induce distortions that are almost invisible, SSL, RoSteALS and StegaStamp produce relatively more visible distortions—local color jitters for SSL, global color jitters for StegaStamp, and global smoothing for RoSteALS. All images presented above are in `uint8` format with value in $[0, 255]$ with resolution $512 \times 512$.

to human eyes. However, existing methods typically work with heuristic penalty or regularization forms of Eq. (2) and therefore do not necessarily find feasible solutions to Eq. (2), e.g., (Zhu et al., 2018). In addition, the choice of $\ell_q$ can differ, e.g., mean squared error (MSE) (Zhu et al., 2018; Zhang et al., 2019a), LPIPS distance (Zhang et al., 2018). Therefore, different methods lead to different levels of visible distortions. Fig. 3 visualizes several popular training-based methods (together with the manually designed DwtDctSVD) and highlights the different distortion levels that they lead to.

**(Blind) image watermarking** Based on steganography, a natural way to trace AI-generated images is to assign *a fixed message $\boldsymbol{w}$* as the signature of the content owner (e.g., a company) and apply steganography to generate images containing the signature, i.e., watermarked images, to achieve (Jiang et al., 2023):

$$D(E(I, \boldsymbol{w})) = \boldsymbol{w}, \quad \forall I \in \mathcal{I}, \quad (\boldsymbol{w} \text{ is a fixed message representing the signature}) \tag{3a}$$

$$D(I) \neq \boldsymbol{w}, \qquad \forall I \in \mathcal{I}, \quad (\text{messages decoded from unwatermarked images should not be } \boldsymbol{w}) \tag{3b}$$

$$E(I, \boldsymbol{w}) \approx I, \qquad \forall I \in \mathcal{I}. \quad (\text{minimal encoding distortion to the image}) \tag{3c}$$

In practice, whether an image $I$ is watermarked by $\boldsymbol{w}$ can be detected by comparing the decoded message with $\boldsymbol{w}$:

$$\mathbb{1}[BA(D(I), \boldsymbol{w}) > \gamma], \tag{4}$$

where $BA$ denotes the bitwise accuracy and $\gamma$ is a preset task-dependent threshold (Jiang et al., 2023; Fernandez et al., 2023; Yu et al., 2021). Due to the similarity between Eq. (3) and Eq. (1), most existing work considers image watermarking as a special application of steganography, e.g., Zhu et al. (2018); Tancik et al. (2020); An et al. (2024); Zhao et al. (2024a). As a result, the learning formulation in Eq. (2) is also widely adopted in works that only focus on watermarking systems, e.g., Zhang et al. (2019b); Fernandez et al. (2023).

The above watermark methods are *post-processing* in nature, as the watermark is embedded on any given image $I$ that is already generated. There is an emerging line of *in-processing* watermark methods that directly modify the image generation process (Zhao et al., 2024a; An et al., 2024), including TreeRing watermark (Wen et al., 2023), stable signature (Fernandez et al., 2023), Gaussian shading watermark (Yang et al., 2024), and pseudorandom error-correcting code watermark Gunn et al. (2024). In these in-processing methods, there is no notion of "clean images" $I$ and every generated watermark image exhibits a semantic shift compared to the image generated without the in-process watermarking; see Fig. 4 for an example generated by the TreeRing watermark.

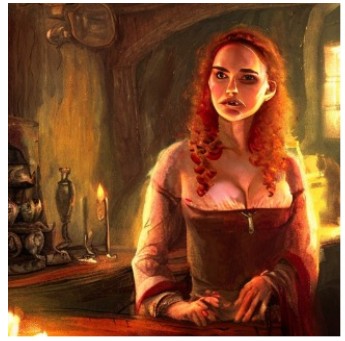 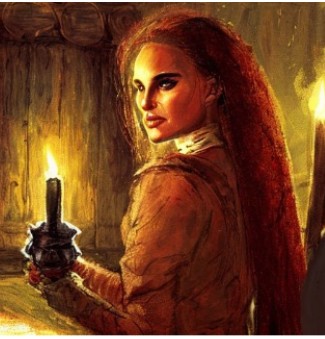

(a) w/o TreeRing watermark   (b) with TreeRing watermark

Figure 4: Visualization of a TreeRing watermark example, where **(a)** and **(b)** are images generated from the same text prompt input using the same image diffusion model without and with the in-process watermarking, respectively.

**In this paper, we focus on post-processing watermark methods due to their flexibility, as they are agnostic to the image generation process**.

**Robustness of watermarking systems**    Robustness of an image watermarking system refers to the extent of the watermark to remain detectable by the decoder $D$ when the watermarked image is manipulated (also called "evaded" if such manipulation is a deliberate attack). Thus, robustness is typically associated with the potential of a watermarking system to be applied to copyright protection and misinformation detection. To stress test the robustness of watermark systems, various watermark evasion techniques have been proposed. These techniques are broadly classified into white-box and black-box evasions, depending on whether any component of the watermark system is known to the evader (An et al., 2024). **In this paper, we focus on black-box evasions where nothing about the watermark system is known**, as in practice, companies tend to keep their watermark system private. Existing black-box evasions can be classified into two groups: **Corruption methods** try to distort watermarked images so that watermarks become corrupted and undetectable. Classical digital editing (e.g., applying Gaussian noise, Gaussian blur, JPEG compression, etc. (Voyatzis & Pitas, 1999)), the query-based adversarial attack (WevadeBQ) in Jiang et al. (2023) and the surrogate attack in Saberi et al. (2023) belong to this group; **Purification methods** treat embedded watermark patterns as noise signals and attempt to remove them using denoising and regeneration techniques, such as BM3D (Dabov et al., 2007), diffusion models (Saberi et al., 2023; Zhao et al., 2024b) and VAE (Zhao et al., 2024b). The rationale behind the purification methods is rooted in Eq. (3), where the original image, if recovered successfully, is always an evasion. In addition, the original image will be the ultimate threat to any watermarking system—the watermark is evaded by an image without any loss of quality.

## 3 Our method: DIP for black-box watermark evasion

### 3.1 Watermark evasion via DIP-based blind denoising

**Deep Image Prior (DIP)**    refers to the technique of using **untrained** DNN as an implicit prior for natural images in solving image recovery problems, **without training on massive data**: for any natural image $I$, DIP parametrizes it as $I = G_{\boldsymbol{\theta}}(\boldsymbol{z})$, where $G_{\boldsymbol{\theta}}$ is typically a trainable convolutional neural network (CNN) and $\boldsymbol{z}$ is a fixed input (typically randomly drawn). Now consider the canonical optimization-based formulation for image recovery problems:

$$\min_I \ \ell(\boldsymbol{y}, f(I)) + \lambda R(I), \tag{5}$$

where $\boldsymbol{y} \approx f(I)$ is the observation model, $\ell(\boldsymbol{y}, f(I))$ measures the recovery error, and $R(\cdot)$ denotes regularization on $I$. DIP transforms the formulation into

$$\min_{\boldsymbol{\theta}} \ \ell(\boldsymbol{y}, f(G_{\boldsymbol{\theta}}(\boldsymbol{z}))) + \lambda R(G_{\boldsymbol{\theta}}(\boldsymbol{z})). \tag{6}$$

---

**Algorithm 1** DIP-based watermark evasion

---

**Require:** A single watermarked image $I_{\boldsymbol{w}}$; a CNN $G_{\boldsymbol{\theta}}(\boldsymbol{z})$ where $\boldsymbol{\theta}$ collects the trainable parameters and $\boldsymbol{z}$ is a fixed vector (randomly drawn as iid Gaussian); a watermark detection API whose input can be an arbitrary image and returns "Yes/No" as the detection output.
1: Randomly initialize $\boldsymbol{\theta}^{(0)}$
2: Solve $\min_{\boldsymbol{\theta}} \ell_2(I_{\boldsymbol{w}}, G_{\boldsymbol{\theta}}(\boldsymbol{z}))$ using the ADAM optimizer for a sufficient number of iterations $N$; record all intermediate results $I_i = G_{\boldsymbol{\theta}^{(i)}}(\boldsymbol{z})$, $\forall i \leq N$, where $i$ denotes the iteration number.
3: Query the detection API using DIP intermediate steps ($I_i$'s); return the $I_i$'s that have no watermark detected.

---

Note that the optimization is *only* with respect to the CNN weights $\boldsymbol{\theta}$. The resulting formulation is then solved by first-order optimizers, such as ADAM (Kingma & Ba, 2014). Such a simple strategy, in combination with appropriate early stopping methods (Li et al., 2021; Wang et al., 2021; Shi et al., 2022b) that pick the best intermediate recovered images, has proved highly successful in solving a wide range of image recovery problems, from simple denoising (Ulyanov et al., 2018) to advanced scientific and medical reconstruction problems (Tirer et al., 2023; Qayyum et al., 2023; Zhuang, 2023; Zhuang et al., 2024; 2023; Li et al., 2024; 2023b;a).

**Watermark evasion via DIP-based blind denoising** Now, consider an arbitrary watermarked image $I_{\boldsymbol{w}} \doteq E(I, \boldsymbol{w})$, where we do not know $E$ or $\boldsymbol{w}$. Since $I_{\boldsymbol{w}} \approx I$ as required in Eq. (3) and $I$ is clearly a successful invasion, it is sensible to try to "purify" or "denoise" $I_{\boldsymbol{w}}$ toward $I$—the intuition behind all purification methods for black-box evasion (Dabov et al., 2007; Saberi et al., 2023; Zhao et al., 2024b).

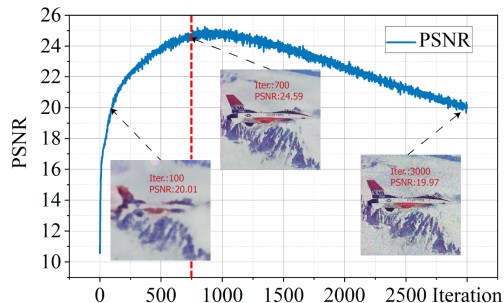

Figure 5: A typical image quality (measured by peak signal-to-noise ratio, PSNR) vs. iteration curve when DIP is used in blind denoising tasks. An early stopping method is used to detect the iteration achieving the peak performance (red dashed line). (Figure adapted from Wang et al. (2021) under the Creative Commons 4.0 license)

DIP has proven effective in single-image denoising, e.g., with Gaussian, impulse, shot noise, etc., when combined with appropriate early stopping strategies (Mataev et al., 2019; Jo et al., 2021; Li et al., 2021; Wang et al., 2021; Li et al., 2023a;b; 2024). In particular, when the noise level is low—which is true for typical watermarking systems as $I_{\boldsymbol{w}}$ is supposed to be very close to $I$, a simple formulation with the standard mean-squared-error (MSE) loss and the additive noise model can perform "blind" simultaneous denoising for multiple types of noise Li et al. (2021); Wang et al. (2021). Inspired by this, we propose a simple DIP-based blind watermark-evasion formulation

$$\min_{\boldsymbol{\theta}} \ \|I_{\boldsymbol{w}} - G_{\boldsymbol{\theta}}(\boldsymbol{z})\|_2^2, \quad (\textbf{DIP-based watermark evasion}) \tag{7}$$

for any given watermarked image $I_{\boldsymbol{w}}$. Unlike DIP-based blind denoising which requires appropriate early stopping strategies to find optimal denoising, we only need to check the evasion success of all iterates when iteratively solving Eq. (7) by querying the watermark decoder. Our entire algorithm pipeline is summarized in Algorithm 1. Although we do not invent DIP-based blind denoising, we are the first to explore it for blind evasion of invisible watermarks. While Rishik (2020) also performs DIP-based watermark removal, it aims at *a different setup*: Rishik (2020) tries to remove visible watermarks by DIP-based image inpainting, which requires the *exact* watermark location as its inpainting mask; in contrast, here we try to remove invisible watermarks using DIP-based denoising, which requires only the watermarked image and nothing else.

## 3.2  Why including DIP-based evasion as a baseline method?

The primary obstacle for purification methods is that the "noise" patterns induced by watermark methods, especially those training-based ones, may not follow any simple noise model. Consequently, classical denoising methods that target specific noise types, such as BM3D, may struggle to remove the watermark. The recent

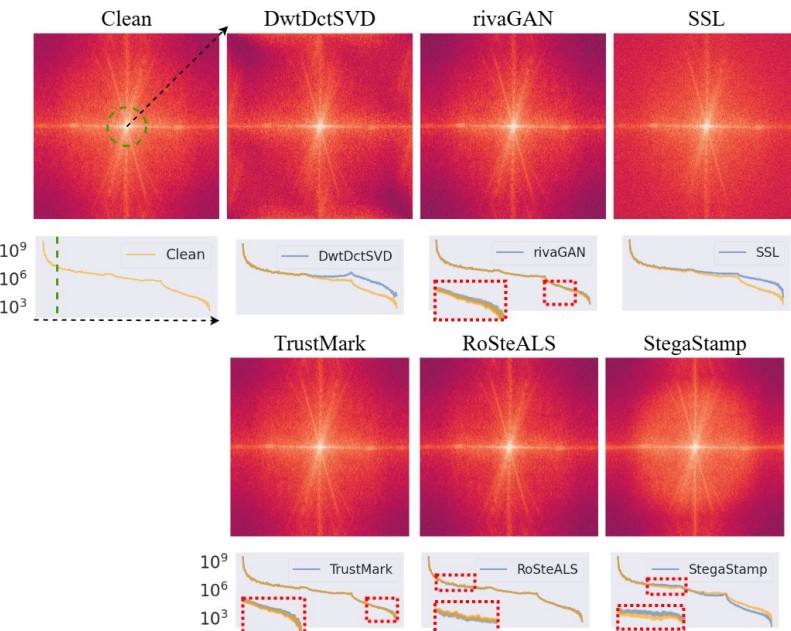

Figure 6: Visualization of 2D Fourier spectra of the clean and watermarked images from Fig. 3 (magnitudes visualized in log scale). The histogram below each spectrum plot shows the band-wise energy distribution in the radial direction (i.e., the dashed arrow direction), where the $y$-axis is in log scale.

regeneration techniques via diffusion models or VAE effectively perform learned denoising. However, a priori, it is unclear they can generalize well to novel watermark patterns. **In contrast, our DIP-based evasion operates on a different principle**: it leverages the different rates at which different frequency components are captured during DIP learning through Eq. (7), which has been consistently observed in prior DIP literature (Ulyanov et al., 2018; Shi et al., 2022a; Li et al., 2021; Wang et al., 2021). Specifically, the $G_{\boldsymbol{\theta}}(\boldsymbol{z})$ term in Eq. (7) picks up the low-frequency components—which dominate natural images, **much faster** than picking up the high-frequency components—which tend to be noise-induced, likely watermark-induced for our case.

- In Fig. 6, we compare the Fourier spectrum of the clean image from Fig. 3 to those of various watermarked versions: clearly, the spectrum of the clean image concentrates on low frequencies, but different watermark methods reshape the Fourier spectrum by mostly changing the mid-to-high frequencies. For example, RoSteALS and StegaStamp mostly affect the low-mid frequencies, DwtDctSVD and SSL mostly focus on mid-high frequencies, and rivaGAN alters high frequencies. The only exception is TrustMark, whose watermark pattern is hard to observe from the frequency spectrum.
- Next, in Fig. 7, we visualize the different learning paces of DIP across different frequency bands, by tracking the frequency band errors (FBEs) similar to Wang et al. (2024); Li et al. (2023a); Zhuang et al. (2024): we first compute the relative per-frequency error in the frequency domain, i.e., $|\mathcal{F}(I_w) - \mathcal{F}(G_{\boldsymbol{\theta}^{(i)}})|/|\mathcal{F}(I_{\boldsymbol{w}})|$, where $\mathcal{F}(\cdot)$ is the discrete Fourier transform, then divide all frequencies into five radial bands from the lowest (1) to the highest (5), and compute the mean errors within each band. It is evident that the lower the frequencies, the faster the FBEs decay.

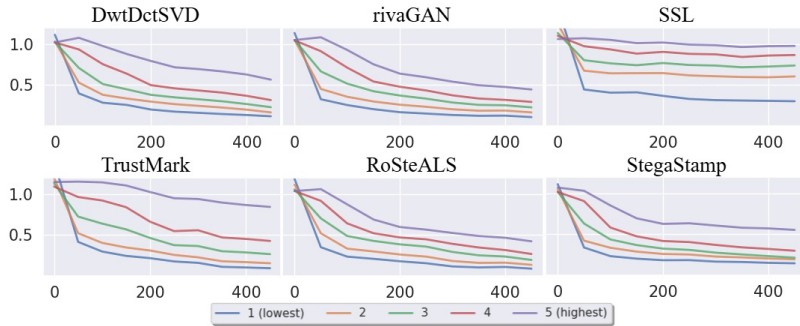

Figure 7: Evolution of the Fourier band errors (FBEs) of DIP's intermediate iterates ($x$-axis: iteration count; $y$-axis: relative band error). We visualize FBEs by dividing all frequency components into five different bands in the radial direction, from the lowest (1) to highest (5).

Such disparate learning paces imply that certain intermediate iterates enjoy sufficient separation of the low-frequency image content and high-frequency watermark patterns, hence the potential for successful evasion. In summary, compared to alternative purification methods that depend on either explicit noise modeling or data-driven priors, our DIP-based evasion relies on frequency separation that is noise-agnostic and distribution-free. Hence, our DIP-based evasion method largely complements the existing ones.

### 3.3 Remarks on DIP-based watermark evasion

Algorithm 1 presents a template algorithm of our DIP-based watermark evasion. Depending on the downstream application, we may need to query a decoder (e.g., a detection API) as stated in Line 3 of Algorithm 1. For example, when the goal is to evade detection while maintaining high image quality, it is essential to query the decoder multiple times to identify the optimal $I_i$. In contrast, for single-shot evasion where image quality is less critical, querying the decoder becomes unnecessary. The consideration applies to all existing black-box evasion methods (e.g., those evaluated in Section 4).

On the other hand, querying the decoder in every intermediate iterate, as described in Algorithm 1, may not be possible, especially since the algorithm often requires up to $\sim 1000$ iterations—for example, the decoder server may impose rate limits on individual user to ensure safety and fairness. It is therefore important to reduce the number of decoder queries and improve query efficiency. One potential approach is to selectively query only high-quality iterates—although the clean image $I$ is unknown, the watermarked image $I_{\boldsymbol{w}}$ is, by design (see Eq. (3)), very close to $I$, and thus can serve as a proxy for estimating the image quality of intermediate iterates; see Appendix D for sample trajectories of image quality and watermark detectability across different watermarks and evasion methods.

Finally, although DIP evasion relies on iterative optimization, it remains computationally efficient in practice. For details on its runtime performance, we refer the reader to Appendix C.

## 4 Qualitative and quantitative evaluation

**Experiment setup** **(1) Datasets**: We use images from two large-scale datasets: **(i)** MS-COCO (Lin et al., 2014) composed of 328K real images and **(ii)** DiffusionDB (Wang et al., 2022) composed of 14 million high-quality AI-generated images. We randomly sample 2000 images from each dataset—the typical scale for the robustness evaluation of watermark systems (An et al., 2024), resize them to $512 \times 512$, and generate images with different watermarks, respectively; **(2) Watermark methods**: We focus on 6 representative and publicly available *post-processing* watermark methods: DwtDctSVD, rivaGAN, SSL, TrustMark, RoSteALS, and StegaStamp, whose watermark patterns vary in the visibility level and Fourier spectrum; see Figs. 3 and 6 and Table 1 for visual and quantitative comparisons, respectively. We also evaluate on the SOTA *in-processing* TreeRing watermark, where 2000 watermarked images are generated using `Gustavosta` stable diffusion prompts from HuggingFace (Santana); **(3) Evasion methods**: In addition to our DIP-based evasion method described in Algorithm 1, [2] we also consider the following classical digital editing methods: **(i)** brightness, **(ii)** contrast, **(iii)** Gaussian noise, **(iv)** JPEG compression, **(v)** bm3d denoising, and recent SOTA purification methods: **(vi)** DiffPure (Saberi et al., 2023), **(vii)** Diffuser (Diffusion attack from (Zhao et al., 2024b)) and **(viii)** VAE regeneration (VAE attack from (Zhao et al., 2024b)) [3].

**Evaluation protocol** As we argue in Section 1 (see also An et al. (2024)), evasion success and image quality are two essential dimensions of watermarking systems. Therefore, we report the *best image quality* each evasion method can achieve while failing watermark detection. To find the "optimal" tradeoff image, we perform an exhaustive search over the allowable ranges of the hyperparameters for each evasion method and look for images with **(i)** watermark undetected and **(ii)** the highest PSNR value with respect to the watermarked image $I_{\boldsymbol{w}}$; see Section 3.3 for justification on why $I_{\boldsymbol{w}}$ is used and Appendix B for details about all hyperparameters. Finally, to quantify the quality of such images, we use three metrics: **(i)** PSNR, **(ii)** Structural Similarity Index Measure (SSIM) (Hore & Ziou, 2010) and **(iii)** 90% quantile of pixel-wise

---

[2]We use the default 'skip' network in the original DIP repo: `https://github.com/DmitryUlyanov/deep-image-prior`.
[3]VAE regeneration using model from Cheng et al. (2020), which gives the best VAE performance as in Jiang et al. (2023).

Table 1: Quantitative visual distortion induced by different watermark methods. We report the mean and standard deviation (in parathesis) of all three quality metrics, calculated over 100 randomly drawn test images. All images are in `uint8` format with value in $[0, 255]$. RivaGAN has the least visible watermarks, while StegaStamp has the most.

| | COCO dataset | | | DiffusionDB dataset | | |
|---|---|---|---|---|---|---|
| Watermark | PSNR ↑ | SSIM ↑ | 90% Quantile ↓ | PSNR ↑ | SSIM ↑ | 90% Quantile ↓ |
| **DwtDctSVD** | 38.31 (2.67) | 0.98 (0.01) | 4.66 (0.90) | 37.88 (3.60) | 0.97 (0.02) | 4.87 (1.56) |
| **rivaGAN** | 40.57 (0.21) | 0.98 (0.01) | 3.00 (0.03) | 40.61 (0.24) | 0.98 (0.01) | 3.00 (0.03) |
| **SSL** | 33.03 (0.23) | 0.88 (0.04) | 7.19 (0.74) | 33.14 (0.55) | 0.88 (0.04) | 7.10 (0.72) |
| **TrustMark** | 40.46 (1.85) | 0.99 (0.01) | 3.84 (0.85) | 40.67 (2.62) | 0.99 (0.01) | 3.84 (1.14) |
| **RoSteALS** | 30.83 (2.71) | 0.95 (0.02) | 12.8 (3.98) | 30.88 (4.12) | 0.95 (0.03) | 12.5 (5.71) |
| **StegaStamp** | 28.34 (1.61) | 0.90 (0.03) | 15.9 (3.18) | 26.58 (2.01) | 0.84 (0.04) | 19.3 (4.96) |

difference, all with respect to the clean image $I$. The quantile metric mainly serves as a supplement, as PSNR and SSIM focus on the average difference while it reflects the difference on the tail—watermark-induced distortion to an image may be highly localized and hence spatially sparse, which might not be captured by averaging metrics; see Fig. 3 and Table 1 for a sense of the visual and quantitative distortions caused by different watermark methods. Since the TreeRing watermark lacks a notion of clean image, we use $I_w$ as the reference in all evaluation experiments related to TreeRing watermark.

**Experiment results** As mentioned in Section 2, the choice of $\gamma$ in the watermark decoder, as shown in Eq. (4), is highly task-dependent. It determines the true positive rate (TPR) and the false positive rate (FPR) in watermark detection—TPR measures the fraction of watermarked images correctly detected, and FPR measures the fraction of clean images wrongly flagged as watermarked images. In general, the higher the $\gamma$ used in the decoder, the lower the true positive rate (TPR) and the false positive rate (FPR). A practically useful watermark decoder should have a TPR close to one and a FPR close to zero. On the other hand, the higher the $\gamma$, the higher the quality of the evasion image can achieve. To account for the effect of $\gamma$ in robustness evaluation, we report in Table 2 the evasion performance of different methods under different $\gamma$'s on COCO images, and Table 3 on DiffusionDB images, together with the TPR/FPR achieved.

From Tables 2 to 4, we observe that: **(1)** The effect of $\gamma$ on TPR/FPR values as well as image quality of different watermarks against the detection threshold agrees with our discussion above. This is true also for the TreeRing watermark, whose detection threshold is based on the $\ell_1$ distance, very different from those for other watermark methods; **(2)** The performance of watermark evasion shown by the column with $\gamma = 0.55$ in Tables 2 and 3 is not quite meaningful—the very high FPR renders the decoder hardly useful in practice; **(3)** In other cases, there is no clear winner among all the evasion methods we evaluate. For example, our DIP-based evasion has the best performance on DwtDctSVD and rivaGAN, and is comparable to the best on SSL; DiffPure is the most effective evasion on TrustMark, RoSteALS and StegaStamp; TreeRing seems most vulnerable to JPEG compression. This highlights the need to include diverse sources of evasion methods for faithful robustness evaluation of watermarking systems, as we argue in Section 3.2.

Moreover, comparing the results of our DIP-based evasion on different watermarks, we observe that evasion images for DwtDctSVD, rivaGAN, and SSL watermarks can achieve very high quality when $\gamma \geq 0.65$, with reasonable TPR/FPR values. This is well expected, as these watermarks mainly cause high-frequency distortions (see Fig. 6), and DIP-based evasion is good at separating high- and low-frequency components and thereby largely removing the watermark during iteration, as argued in Section 3.2. In contrast, DIP-based evasion shows limited performance on TrustMark, RoSteALS, and StegaStamp. This is because these watermark methods induce substantial mid- and low-frequency distortions, which DIP picks up in early stages and so are hard to separate from the clean image content.

To better understand this, we compute the average relative FBE across 10 radial frequency bands using 100 images watermarked by each method. This is done by partitioning the Fourier spectrum of the watermark pattern $(I_w - I)$ into different radial bands—a quantitative analysis of qualitative results in Fig. 6. As shown in Table 5, Trustmark, RoSteALS, and StegaStamp exhibit significantly higher FBE in the lower mid-frequency bands (Bands 3–5) compared to DwtDctSVD, rivaGAN, and SSL. Notably, comparing SSL

Table 2: The best image quality produced by different evasion methods under different detection threshold $\gamma$ on the COCO dataset. Here, we report the mean value of PSNR-SSIM-90% Quantile (Q.). We highlight the best evasion method under each watermark and $\gamma$ in **boldface**. For fair comparison, we mask out cases where one evasion method cannot evade $\geq 90\%$ of the watermarked images.

| COCO dataset | | $\gamma = 0.55$ | $\gamma = 0.65$ | $\gamma = 0.75$ | $\gamma = 0.85$ |
|---|---|---|---|---|---|
| **DwtDctSVD** | TPR↑ / FPR↓ | 1.00 / 0.23 | 0.99 / 0.03 | 0.99 / 0.01 | 0.99 / 0.00 |
| brightness | PSNR - SSIM - Q. | 19.22 - 0.88 - 52.6 | 20.00 - 0.90 - 47.5 | 20.67 - 0.91 - 43.4 | 21.52 - 0.93 - 39.0 |
| contrast | | 25.27 - 0.89 - 26.7 | 26.02 - 0.91 - 24.1 | 26.67 - 0.91 - 22.0 | 27.52 - 0.92 - 19.7 |
| Gaussian noise | | 16.30 - 0.19 - 66.1 | 17.54 - 0.23 - 56.8 | 18.62 - 0.26 - 50.3 | 20.05 - 0.31 - 42.3 |
| JPEG | | 31.52 - 0.88 - 11.2 | 31.74 - 0.89 - 10.9 | 31.89 - 0.89 - 10.7 | 32.13 - 0.90 - 10.4 |
| bm3d | | **** | **** | ** | 30.26 - 0.87 - 9.6 |
| DiffPure | | 28.20 - 0.79 - 17.3 | 29.61 - 0.83 - 14.2 | 29.90 - 0.84 - 13.6 | 29.94 - 0.84 - 13.6 |
| Diffuser | | ** | ** | *25.87 - 0.74 - 20.7 | 26.61 - 0.76 - 19.2 |
| VAE | | **** | *32.38 - 0.88 - 10.5 | 33.47 - 0.90 - 9.2 | 34.51 - 0.92 - 8.1 |
| DIP (ours) | | **34.87 - 0.96 - 7.2** | **35.50 - 0.96 - 6.7** | **35.85 - 0.96 - 6.4** | **36.22 - 0.97 - 6.1** |
| **rivaGAN** | TPR↑ / FPR↓ | 0.99 / 0.25 | 0.99 / 0.03 | 0.99 / 0.01 | 0.99 / 0.00 |
| brightness | PSNR - SSIM - Q. | **** | 6.91 - 0.12 - 181 | 7.31 - 0.17 - 173 | 8.02 - 0.27 - 159 |
| contrast | | **** | *13.16 - 0.50 - 90.0 | 13.53 - 0.53 - 86.1 | 14.20 - 0.58 - 79.8 |
| Gaussian noise | | 11.06 - 0.07 - 121 | 13.07 - 0.11 - 96.3 | 14.61 - 0.14 - 80.7 | 16.42 - 0.19 - 65.7 |
| JPEG | | **** | ** | 28.22 - 0.79 - 16.7 | 30.16 - 0.85 - 13.2 |
| bm3d | | **** | **** | **** | **** |
| DiffPure | | *26.51 - 0.73 - 21.7 | 28.84 - 0.80 - 15.9 | 29.72 - 0.83 - 14.1 | 29.96 - 0.84 - 13.5 |
| Diffuser | | **** | *25.40 - 0.72 - 22.1 | *26.28 - 0.74 - 20.0 | 26.78 - 0.75 - 18.9 |
| VAE | | ** | *32.34 - 0.88 - **10.3** | 33.28 - 0.90 - 9.3 | 34.21 - 0.91 - 8.3 |
| DIP (ours) | | **29.87 - 0.87 - 17.8** | **32.64 - 0.92 - 11.1** | **34.02 - 0.94 - 8.9** | **35.20 - 0.95 - 7.4** |
| **SSL** | TPR↑ / FPR↓ | 1.00 / 0.33 | 1.00 / 0.07 | 0.99 / 0.01 | 0.99 / 0.00 |
| brightness | PSNR - SSIM - Q. | **** | **** | **** | *13.92 - 0.44 - 149 |
| contrast | | **** | **** | ** | *19.12 - 0.67 - 58.6 |
| Gaussian noise | | *18.89 - 0.28 - 51.4 | 21.67 - 0.39 - 36.7 | 23.98 - 0.48 - 27.8 | 25.42 - 0.54 - 22.7 |
| JPEG | | *28.18 - 0.79 - 16.7 | 30.05 - 0.84 - 13.4 | 31.28 - 0.86 - 11.4 | 32.73 - 0.89 - 9.5 |
| bm3d | | *27.99 - 0.79 - 14.4 | 29.59 - 0.84 - 11.7 | 30.79 - 0.87 - **9.6** | 31.18 - 0.89 - 8.9 |
| DiffPure | | 27.18 - 0.75 - 19.7 | 28.85 - 0.80 - 15.6 | 29.32 - 0.82 - 14.6 | 29.38 - 0.82 - 14.4 |
| Diffuser | | ** | *25.39 - 0.68 - 22.1 | *25.80 - 0.69 - 21.1 | *25.90 - 0.69 - 20.8 |
| VAE | | ***31.21 - 0.85 - 11.9** | ***32.03 - 0.88 - 10.8** | 32.83 - 0.90 - 9.8 | 33.50 - 0.91 - 9.0 |
| DIP (ours) | | 23.73 - 0.76 - 31.4 | 28.21 - 0.85 - 17.4 | 31.46 - **0.90** - 11.2 | **33.64 - 0.92 - 8.4** |
| **TrustMark** | TPR↑ / FPR↓ | 1.00 / 0.13 | 1.00 / 0.00 | 1.00 / 0.00 | 1.00 / 0.00 |
| brightness | PSNR - SSIM - Q. | **** | ** | *6.86 - 0.11 - 182 | *7.27 - 0.17 - 173 |
| contrast | | **** | ** | *13.31 - 0.54 - 88.3 | *13.84 - 0.58 - 88.2 |
| Gaussian noise | | 9.57 - 0.05 - 144 | 11.17 - 0.07 - 119 | 12.72 - 0.09 - 99.0 | 14.49 - 0.13 - 80.9 |
| JPEG | | **** | **** | *25.18 - 0.70 - 22.9 | *26.35 - 0.73 - 20.4 |
| bm3d | | —— | —— | —— | —— |
| DiffPure | | **26.34 - 0.73 - 20.4** | **27.81 - 0.77 - 17.3** | **29.09 - 0.80 - 15.3** | **30.22 - 0.84 - 13.2** |
| Diffuser | | **** | *25.62 - 0.74 - 21.1 | *26.19 - 0.75 - 19.7 | 26.70 - 0.77 - 18.5 |
| VAE | | —— | —— | **** | **** |
| DIP (ours) | | ** | *13.64 - 0.48 - 91.1 | 15.88 - 0.56 - 73.4 | 18.97 - 0.66 - 52.8 |
| **RoSteALS** | TPR↑ / FPR↓ | 1.00 / 0.29 | 1.00 / 0.01 | 1.00 / 0.00 | 1.00 / 0.00 |
| brightness | PSNR - SSIM - Q. | —— | —— | —— | —— |
| contrast | | —— | —— | —— | —— |
| Gaussian noise | | —— | **** | **** | *9.53 - 0.04 - 145 |
| JPEG | | —— | —— | —— | **** |
| bm3d | | —— | —— | —— | —— |
| DiffPure | | ***19.12 - 0.49 - 50.6** | **22.98 - 0.62 - 30.7** | **24.91 - 0.69 - 24.3** | **26.28 - 0.74 - 20.6** |
| Diffuser | | —— | —— | —— | **** |
| VAE | | —— | —— | —— | —— |
| DIP (ours) | | **** | *11.08 - 0.38 - 113 | 12.09 - 0.42 - 105 | 16.36 - 0.56 - 73.5 |
| **StegaStamp** | TPR↑ / FPR↓ | 1.00 / 0.18 | 1.00 / 0.01 | 1.00 / 0.00 | 1.00 / 0.00 |
| brightness | PSNR - SSIM - Q. | —— | —— | —— | —— |
| contrast | | —— | —— | **** | *13.22 - 0.52 - 88.9 |
| Gaussian noise | | **** | *7.83 - 0.03 - 175 | 9.26 - 0.05 - 148 | 11.24 - 0.08 - 118 |
| JPEG | | **** | **** | **** | ** |
| bm3d | | —— | —— | —— | —— |
| DiffPure | | **19.57 - 0.50 - 45.6** | **22.42 - 0.61 - 31.5** | **23.81 - 0.66 - 26.7** | **24.67 - 0.70 - 24.1** |
| Diffuser | | —— | —— | **** | ** |
| VAE | | —— | —— | —— | —— |
| DIP (ours) | | ** | *12.29 - 0.40 - 102 | 14.00 - 0.47 - 87.5 | 16.05 - 0.55 - 70.9 |

The following markers are used for the purpose of fair comparison of the best evasion image quality:

—— Evasion method only successfully evade $< 10\%$ of the watermarked images.

**** Evasion method only successfully evade $< 75\%$ of the watermarked images.

** Evasion method only successfully evade $< 90\%$ of the watermarked images.

* Evasion method successfully evade $\geq 90\%$ of the watermarked images, but $< 100\%$.

Table 3: The best image quality produced by different evasion methods under different detection threshold $\gamma$ on the DiffusionDB dataset. Here, we report the mean value of PSNR-SSIM-90% Quantile (Q.). We highlight the best evasion method under each watermark and $\gamma$ in **boldface**. For fair comparison, we mask out cases where one evasion method cannot evade $\geq 90\%$ of the watermarked images.

| DiffusionDB dataset | | $\gamma = 0.55$ | $\gamma = 0.65$ | $\gamma = 0.75$ | $\gamma = 0.85$ |
|---|---|---|---|---|---|
| **DwtDctSVD** | TPR↑ / FPR↓ | 1.00 / 0.23 | 0.99 / 0.03 | 0.99 / 0.01 | 0.99 / 0.00 |
| brightness | PSNR - SSIM - Q. | 21.25 - 0.90 - 43.0 | 21.94 - 0.91 - 39.1 | 22.54 - 0.92 - 35.9 | 23.34 - 0.93 - 32.3 |
| contrast | | 26.89 - 0.91 - 22.2 | 27.56 - 0.92 - 20.1 | 28.11 - 0.93 - 18.6 | 28.85 - 0.94 - 16.8 |
| Gaussian noise | | 16.52 - 0.22 - 64.3 | 17.73 - 0.25 - 55.5 | 18.81 - 0.29 - 49.1 | 20.07 - 0.33 - 42.0 |
| JPEG | | 30.47 - 0.86 - 13.2 | 30.81 - 0.87 - 12.7 | 31.00 - 0.88 - 12.4 | 31.30 - 0.88 - 12.0 |
| bm3d | | **** | **** | ** | *28.94 - 0.85 - 11.6 |
| DiffPure | | 27.69 - 0.78 - 18.8 | 29.09 - 0.82 - 15.8 | 29.52 - 0.83 - 15.0 | 29.56 - 0.83 - 14.9 |
| Diffuser | | ** | ** | *25.88 - 0.75 - 21.7 | *26.54 - 0.77 - 20.5 |
| VAE | | **** | *31.91 - 0.87 - 11.5 | 32.90 - 0.89 - 10.3 | 32.92 - 0.91 - 9.0 |
| DIP (ours) | | **35.29 - 0.96 - 6.9** | **35.91 - 0.96 - 6.4** | **36.25 - 0.97 - 6.1** | **36.53 - 0.97 - 5.9** |
| **rivaGAN** | TPR↑ / FPR↓ | 0.99 / 0.25 | 0.99 / 0.03 | 0.99 / 0.01 | 0.99 / 0.00 |
| brightness | PSNR - SSIM - Q. | **** | *7.38 - 0.13 - 173 | 7.83 - 0.19 - 165 | 8.70 - 0.31 - 150 |
| contrast | | **** | *13.47 - 0.51 - 86.8 | 13.93 - 0.55 - 82.5 | 14.73 - 0.60 - 75.5 |
| Gaussian noise | | 10.96 - 0.08 - 123 | 12.98 - 0.12 - 97.3 | 14.55 - 0.16 - 81.4 | 16.56 - 0.21 - 64.8 |
| JPEG | | **** | *26.31 - 0.74 - 20.9 | *27.84 - 0.79 - 17.5 | 29.61 - 0.84 - 14.1 |
| bm3d | | **** | **** | **** | **** |
| DiffPure | | 26.11 - 0.73 - 22.9 | 28.25 - 0.79 - 16.8 | 28.88 - 0.81 - 15.4 | 29.02 - 0.82 - 15.1 |
| Diffuser | | **** | *25.43 - 0.73 - 22.5 | *26.12 - 0.75 - 20.8 | 26.47 - 0.76 - 20.0 |
| VAE | | ** | *32.29 - 0.88 - 10.4 | *33.13 - 0.90 - 9.4 | 33.94 - 0.91 - 8.5 |
| DIP (ours) | | **31.91 - 0.89 - 14.5** | **34.48 - 0.93 -** 9.1 | **35.72 - 0.95 - 7.4** | **36.75 - 0.96 - 6.2** |
| **SSL** | TPR↑ / FPR↓ | 1.00 / 0.33 | 1.00 / 0.07 | 0.99 / 0.01 | 0.99 / 0.00 |
| brightness | PSNR - SSIM - Q. | **** | **** | ** | *16.91 - 0.53 - 97.9 |
| contrast | | **** | **** | ** | *21.54 - 0.73 - 47.9 |
| Gaussian noise | | *17.43 - 0.26 - 63.8 | 20.90 - 0.38 - 41.0 | 23.48 - 0.48 - 29.7 | 25.31 - 0.56 - 23.0 |
| JPEG | | *27.07 - 0.77 - 19.4 | 28.97 - 0.82 - 15.4 | 30.47 - 0.85 - 12.7 | 32.39 - 0.89 - 9.9 |
| bm3d | | *26.69 - 0.77 - 16.3 | 28.00 - 0.82 - 13.3 | 29.00 - 0.85 - 11.1 | 29.25 - 0.86 - 10.5 |
| DiffPure | | 26.67 - 0.75 - 21.0 | 28.05 - 0.79 - 17.3 | 28.40 - 0.80 - 16.3 | 28.43 - 0.80 - 16.2 |
| Diffuser | | ** | *25.00 - 0.68 - 23.6 | 25.43 - 0.69 - 22.5 | 25.49 - 0.70 - 22.4 |
| VAE | | ***30.26 - 0.85 - 12.3** | ***31.79 - 0.88 - 11.0** | **32.43 - 0.90 - 9.9** | 33.01 - 0.91 - 9.2 |
| DIP (ours) | | 23.14 - 0.72 - 35.2 | 27.95 - 0.84 - 18.2 | 31.27 - 0.89 - 11.8 | **33.84 - 0.92 - 8.2** |
| **TrustMark** | TPR↑ / FPR↓ | 1.00 / 0.13 | 1.00 / 0.00 | 1.00 / 0.00 | 1.00 / 0.00 |
| brightness | PSNR - SSIM - Q. | **** | ** | *7.09 - 0.10 - 179 | *7.43 - 0.15 - 172 |
| contrast | | **** | ** | *13.52 - 0.55 - 87.3 | *13.98 - 0.59 - 83.2 |
| Gaussian noise | | *9.62 - 0.05 - 143 | 11.17 - 0.08 - 119 | 12.67 - 0.10 - 100 | 14.45 - 0.14 - 81.7 |
| JPEG | | **** | **** | *25.06 - 0.69 - 24.5 | *26.08 - 0.72 - 22.4 |
| bm3d | | —— | | | |
| DiffPure | | **26.60 - 0.73 - 21.8** | **27.91 - 0.77 - 18.7** | **28.99 - 0.79 - 16.6** | **30.07 - 0.83 - 14.4** |
| Diffuser | | ** | *25.79 - 0.74 - 21.9 | *26.40 - 0.76 - 20.4 | 26.96 - 0.77 - 19.2 |
| VAE | | —— | —— | **** | **** |
| DIP (ours) | | ** | *14.16 - 0.5 - 88.4 | 16.19 - 0.57 - 73.0 | 19.07 - 0.65 - 54.6 |
| **RoSteALS** | TPR↑ / FPR↓ | 1.00 / 0.29 | 1.00 / 0.01 | 1.00 / 0.00 | 1.00 / 0.00 |
| brightness | PSNR - SSIM - Q. | —— | —— | —— | —— |
| contrast | | —— | —— | —— | —— |
| Gaussian noise | | —— | **** | **** | *9.92 - 0.05 - 139 |
| JPEG | | —— | —— | —— | **** |
| bm3d | | —— | —— | —— | —— |
| DiffPure | | **20.86 - 0.57 - 43.7** | **24.06 - 0.66 - 29.3** | **25.74 - 0.72 - 23.8** | **26.93 - 0.76 - 20.6** |
| Diffuser | | —— | —— | —— | **** |
| VAE | | —— | —— | —— | —— |
| DIP (ours) | | ** | *11.67 - 0.42 - 108 | 12.83 - 0.46 - 98.9 | 17.47 - 0.60 - 67.0 |
| **StegaStamp** | TPR↑ / FPR↓ | 1.00 / 0.18 | 1.00 / 0.01 | 1.00 / 0.00 | 1.00 / 0.00 |
| brightness | PSNR - SSIM - Q. | —— | —— | —— | —— |
| contrast | | —— | —— | **** | ** |
| Gaussian noise | | **** | *8.03 - 0.04 - 170 | 9.49 - 0.06 - 145 | 11.66 - 0.09 - 113 |
| JPEG | | **** | **** | **** | ** |
| bm3d | | —— | —— | —— | —— |
| DiffPure | | *19.70 - 0.54 - 45.6 | **22.42 - 0.63 - 32.2** | **23.73 - 0.68 - 27.5** | **24.55 - 0.71 - 24.9** |
| Diffuser | | —— | —— | **** | *21.97 - 0.61 - 32.4 |
| VAE | | —— | —— | —— | —— |
| DIP (ours) | | ** | *13.01 - 0.44 - 96.7 | 14.69 - 0.50 - 82.7 | 16.54 - 0.57 - 68.5 |

The following markers are used for the purpose of fair comparison of the best evasion image quality:

—— Evasion method only successfully evade < 10% of the watermarked images.

**** Evasion method only successfully evade < 75% of the watermarked images.

**     Evasion method only successfully evade < 90% of the watermarked images.

*       Evasion method successfully evade $\geq 90\%$ of the watermarked images, but < 100%.

Table 4: The best image quality produced by different evasion methods under different detection threshold on Tree-Ring watermarked images. Here, we report the mean value of PSNR-SSIM-90% Quantile (Q.). We highlight the best number under each watermark and threshold. Note that TreeRing relies on thresholding the $\ell_1$ distance for pattern matching, which is different from $\gamma$ used in other watermarks. For fair comparison, we mask out cases where one evasion method cannot evade $\geq 90\%$ of the watermarked images.

| Tree-Ring | Threshold | 70 | 60 | 50 | 40 |
|---|---|---|---|---|---|
| | TPR↑ / FPR↓ | 1.00 / 0.01 | 1.00 / 0.00 | 0.99 / 0.00 | 0.95 / 0.00 |
| Gaussian noise | PSNR - SSIM - Q. | ** | 16.49 - 0.19 - 70.6 | 23.06 - 0.42 - 32.9 | 25.75 - 0.53 - 22.3 |
| JPEG | | —— | *28.45 - 0.78 - 17.2 | 31.37 - 0.86 - 12.3 | 34.64 - 0.92 - 8.4 |
| bm3d | | **** | *26.40 - **0.79 - 14.6** | 28.66 - 0.85 - 10.2 | 29.47 - 0.88 - 8.5 |
| DiffPure | | **23.20 - 0.63 - 31.8** | 27.79 - 0.77 - 18.7 | 29.79 - 0.83 - 14.1 | 30.16 - 0.84 - 13.0 |
| Diffuser | | —— | **** | ** | *28.01 - 0.84 - 16.1 |
| VAE | | —— | **** | *34.06 - 0.89 - 9.0 | *35.58 - 0.93 - 7.2 |
| DIP (ours) | | *12.28 - 0.42 - 102.7 | 17.67 - 0.59 - 64.4 | 26.14 - 0.79 - 26.1 | 34.11 - 0.93 - 9.6 |

The following markers are used for the purpose of fair comparison of the best evasion image quality:

—— Evasion method only successfully evade $< 10\%$ of the watermarked images.
**** Evasion method only successfully evade $< 75\%$ of the watermarked images.
** Evasion method only successfully evade $< 90\%$ of the watermarked images.
* Evasion method successfully evade $\geq 90\%$ of the watermarked images, but $< 100\%$.

| Watermarks | (Low) Band 1 | Band 2 | Band 3 | Band 4 | Band 5 | Band 6 | Band 7 | Band 8 | Band 9 | Band 10 (High) |
|---|---|---|---|---|---|---|---|---|---|---|
| dwtDctSvd | 0.025 | 0.044 | 0.059 | 0.074 | 0.091 | 0.103 | 0.109 | 0.109 | 0.110 | 0.371 |
| rivaGAN | 0.016 | 0.033 | 0.061 | 0.097 | 0.132 | 0.163 | 0.196 | 0.214 | 0.237 | 0.240 |
| SSL | 0.011 | 0.036 | 0.070 | 0.119 | 0.174 | 0.222 | 0.289 | 0.331 | 0.377 | 0.765 |
| Trustmark | 0.017 | 0.075 | **0.177** | **0.187** | **0.189** | 0.175 | 0.148 | 0.113 | 0.094 | 0.109 |
| Rosteals | 0.085 | 0.282 | **0.386** | **0.315** | **0.411** | 0.436 | 0.475 | 0.556 | 0.620 | 0.307 |
| StegaStamp | 0.166 | 0.350 | **0.432** | **0.443** | **0.492** | 0.518 | 0.503 | 0.481 | 0.434 | 1.768 |

Table 5: Average relative Fourier Band Error (FBE) across 10 radial frequency bands for different watermarking methods. Values in bold indicate watermarks with notably high energy in the mid-frequency bands. SSL and Trustmark are also highlighted for comparison: SSL exhibits higher energy in high-frequency components but is vulnerable to DIP evasion, whereas Trustmark shows stronger mid-frequency energy and is more resistant to DIP evasion.

(whose FBE has larger magnitudes on Band 6-10) and Trustmark (whose FBE has larger magnitudes on Band 3-5), it is even clearer that leveraging lower frequency bands is the key to counter DIP evasions.

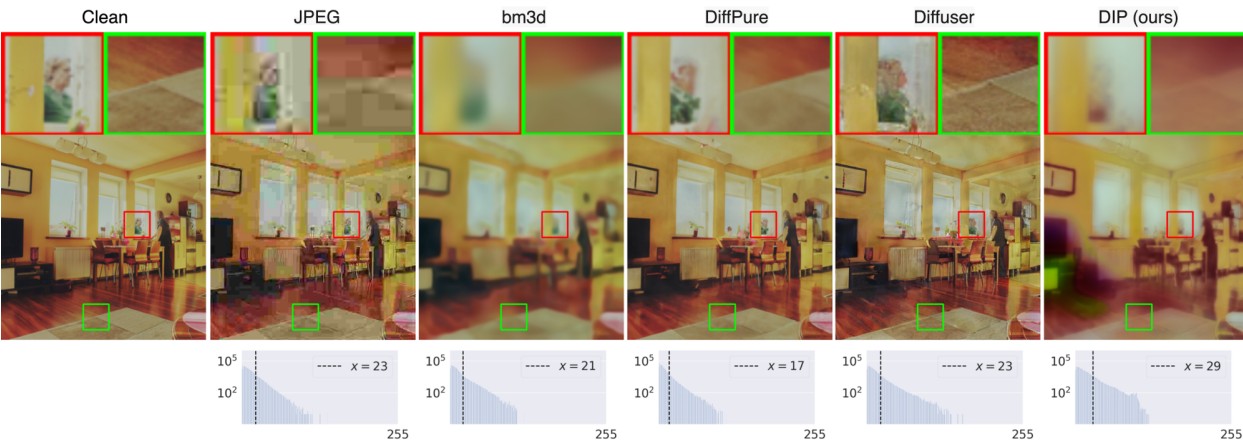

Figure 8: Visualization of the evasion images found by different evasion methods on a StegaStamp watermarked image (with $\gamma = 0.75$; top row) and the respective histograms of the pixel difference ($y-$axis in log scale) between the evasion image and the clean image (bottom row). The vertical dashed line marks the 90% quantile.

## 5 Conclusion and discussion

With the results and the analysis above, we can conclude that there is *no universal best evasion methods* for existing watermarking systems. In general, our DIP-based evasion is most effective in evading invisible watermarks that induce high-frequency distortions (e.g., DwtDctSVD, rivaGAN and SSL), and is partially successful in evading *in-processing* watermarks such as TreeRing. Its limited performance for RoSteALS and StegaStamp implies that exploiting low- and mid-frequency distortions is a viable way for watermarking systems to counteract our DIP-based evasion. Also, for these watermark methods, the regeneration evasion DiffPure has proved effective.

Moreover, for relatively visible watermarks (e.g., StegaStamp), the evasion images generated by all evasion methods always contain visible artifacts; see Fig. 8 for an example. Therefore, the future of learning-based watermarks is not all pessimistic: they may not be reliable for copyright protection, but may be promising in misinformation prevention. This is because of the distinct requirements of these two kinds of applications: for copyright protection, watermarks are expected to remain detectable as long as the image content is recognizable even under severe corruptions due to evasion—which may be too hard to achieve. In contrast, to prevent misinformation, it might be sufficient to achieve either of the following to mitigate the harm: **(i)** the watermark patterns can be detected by eyes, e.g., an overlaid logo or unnatural perturbations such as Figs. 1 and 8, raising suspicion that the image is already manipulated or fake; **(ii)** the watermark can be detected by an algorithmic decoder. For this purpose, watermarks such as StegaStamp may be sufficient.

## 6 Ethical statement

One potential ethical concern regarding this paper is that it could facilitate the unauthorized removal of watermarks, thereby enabling copyright infringement. However, this concern may be unfounded: Tutorials on using Deep Image Prior (DIP) to remove visible watermarks (e.g., Rishik (2020)) are already publicly available; While research on invisible watermarking is active, such methods have not yet been widely deployed in real-world applications. Rather than promoting misuse, our work aims to proactively identify limitations in current invisible watermarking techniques and to offer concrete recommendations for developing more robust watermarking strategies.

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

## A  DALLE-2 visible watermark

An example of a visible `DALLE-2` watermark in colored blocks is shown Fig. 9.

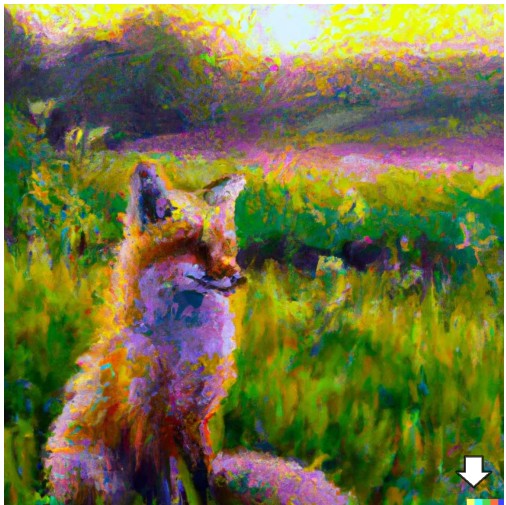

Figure 9: An example image generated by `DALLE-2` from the official website: `https://openai.com/index/dall-e-2/`, where the color code watermark is visible at the bottom-right corner.

Table 6: Details of hyperparameters used in our exhaustive search

| Evasion method | Hyperparameter | Search range | Search resolution | Common default value |
|---|---|---|---|---|
| **brightness** | Enhancement factor | [0.01, 1] | 0.01 | 0.5 |
| **contrast** | Enhancement factor | [0.01, 1] | 0.01 | 0.5 |
| **Gaussian Noise** | Standard deviation | [0.01, 1] | 0.01 | 0.1 |
| **JPEG** | Quality factor | [1, 100] | 1 | 50 |
| **bm3d** | Noise standard deviation | [0.1, 5] | 0.05 | 0.1 |
| **DiffPure** | Diffusion noise level | [0.1, 1] | 0.1 | 0.1 - 0.3 |
| **Diffuser** | Diffusion inverse steps | [10, 100] | 10 | 60 |
| **VAE-Cheng2020** | VAE compression quality indenx | [1, 6] | 1 | 3 |
| **DIP** (ours) | Number of iteration | [1, 500] | 10 | - |

## B    Ranges of hyperparameters of various evasion methods

Table 6 presents the range of each hyperparameter and the grid resolution in our exhaustive search. If we simply use these typical values without exhaustive search, it is very likely that we will overestimate the level of robustness and underestimate the evasion image quality. Fig. 10 shows an example of the exhaustive search process on our DIP-based evasion.

In Table 6, the "quality index" for the VAE refers to the selection of a specific pretrained model (among the six provided in the original work), hence the minimal grid resolution for this hyperparameter is 1. The final column lists the default values commonly used in the literature for robustness evaluation (e.g., Zhao et al. (2024b); Saberi et al. (2023)). To the best of our knowledge, the rationale behind the choice of these default settings is not well documented.

## C    Runtime comparison of different watermark evasions

Table 7: Runtime comparison of evading a single watermarked image by different watermark evasion methods

| Method | Evasion configuration | Time (s) |
|---|---|---|
| Diffuser | 10 different diffusion steps | 3.85 |
| BM3D | 5 different standard deviation parameters | 18.2 |
| DIP | 500 iterations | 26.6 |
| DiffPure | 10 different diffusion steps | 290.75 |

We perform a runtime comparison of different watermark evasion methods mentioned in this paper. The experiment is carried out on a local desktop with `Windows OS` equipped with an `Intel Core i7-12700K` processor and an `NVIDIA RTX 3080 GPU`, on the evasions of a single image. The result can be found in Table 7: Our DIP-based evasion is comparable to Diffuser and BM3D, and is substantially faster than DiffPure. Moreover, we note that recent research has tried to accelerate DIP-based denoising, e.g., Li et al. (2023a), which—although beyond the scope of this paper—represents promising directions for further improving the computational efficiency of our DIP-based watermark evasion.

## D    Image quality v.s. evasion success

Fig. 10 presents several illustrative trajectories of the relationship between image quality and the evasion success evaluated on our DIP-based evasion.

## E    Additional visualization of evasion image quality on rivaGAN

Fig. 11 presents evasion images with the best visual quality from the different evasion methods considered in this paper. Note that all other methods except for DIP results in visual artifacts in the evasion image, e.g., pixel jitters (Gaussian Noise, JPEG and Diffuser) or overly smoothing effect (bm3d, DiffPure and VAE).

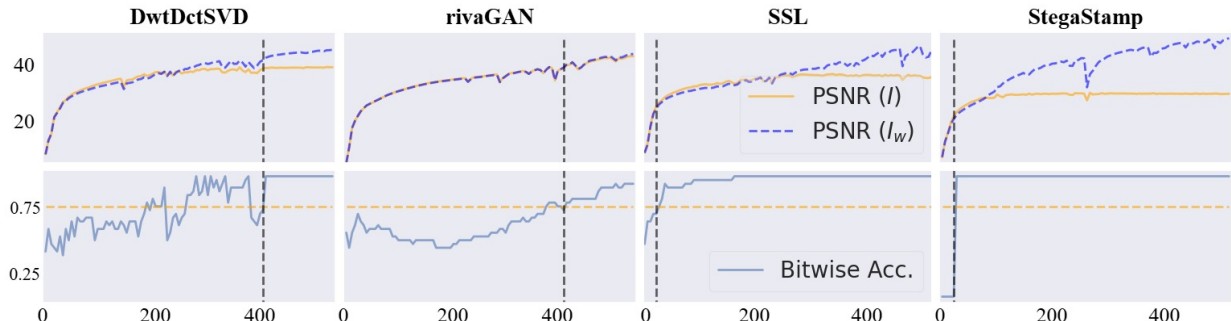

Figure 10: The PSNR trajectories of DIP-based evasion (top row) with respect to the watermarked image (in blue) and with respect to the original image (orange), and the corresponding trajectory of evasion performance (bottom row, measured by $BA$), for different watermark systems. Consider the BA threshold $\gamma = 0.75$ for detection (marked by horizontal orange lines). The vertical black lines mark the best-quality evasion images.

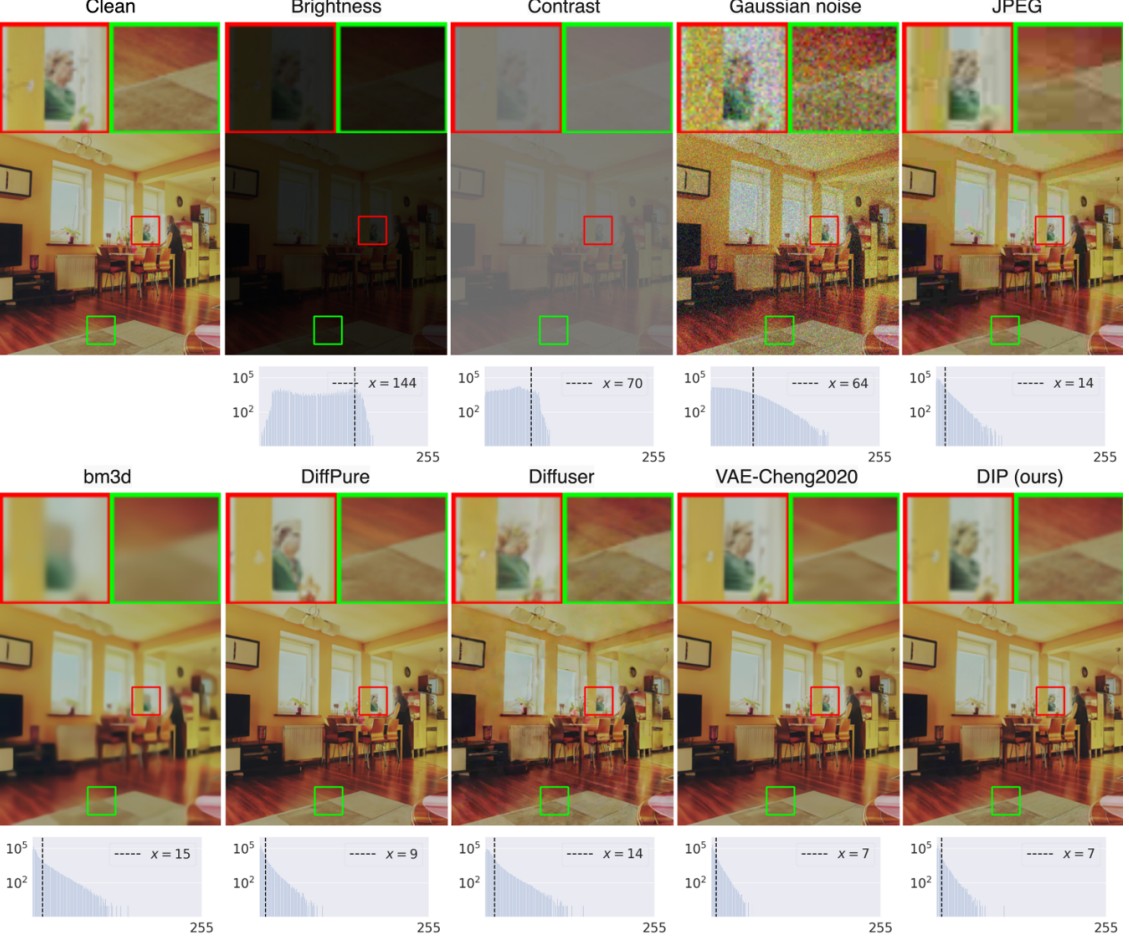

Figure 11: Visualization of the evasion images found by different evasion methods on a rivaGAN watermarked image (with $\gamma = 0.75$; top row) and the respective histograms of the pixel difference ($y-$axis in log scale) between the evasion image and the clean image (bottom row). The vertical dashed line marks the 90% quantile. We can observe that the evasion image produced by DIP has almost no loss of image quality.

## F   Additional discussion on the WIND watermark

Arabi et al. (2024) proposes an inpainting-based variant of the WIND watermark (WIND$_{\text{inpainting}}$), whose underlying principle is similar to that of TreeRing (Wen et al., 2023) and aims to extend diffusion-based in-processing watermarking methods to post-processing scenarios. The primary focus of Arabi et al. (2024) is the identification (i.e., recovery) of the watermark key, but not watermark detection—the binary classification of watermarked vs. non-watermarked images; see the evaluation in Table 1 in Wen et al. (2023).

To evaluate the detection performance of WIND$_{\text{inpainting}}$, we perform the following experiment: We sample 100 images from the COCO dataset and generate the corresponding watermarked versions using WIND$_{\text{inpainting}}$ (Arabi et al., 2024). We then apply the WIND$_{\text{inpainting}}$ decoder to all 200 images (clean and watermarked) and record the first-stage distance for each. As described in Arabi et al. (2024), this distance serves as the basis for watermark detection. We plot the histogram of the first-stage distance for clean and watermarked images, respectively, as shown in Fig. 12. We observe that thresholding the first-stage distance produced by

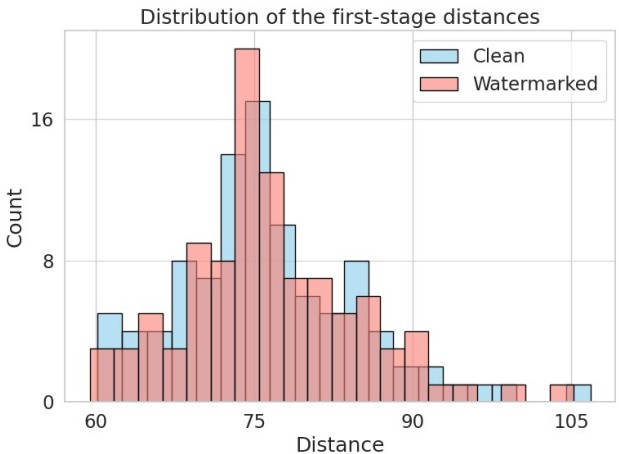

Figure 12: Distribution of the first-stage distance for clean and watermarked images decoded by WIND$_{\text{inpainting}}$

the WIND$_{\text{inpainting}}$ decoder is insufficient for reliably distinguishing between clean and watermarked images. Consequently, we do not include WIND$_{\text{inpainting}}$ in our comparison.

