# OpenReview forum: "A Baseline Method for Removing Invisible Image Watermarks using Deep Image Prior"
_TMLR — Accepted by TMLR_

### Review · Reviewer_FSQv · 2025-04-04

**Summary Of Contributions:**

The author presents a Deep Image Prior (DIP) technique for watermark removal that requires no access to additional information. This method demonstrates superior effectiveness against certain types of watermarks when compared to other evaluated approaches. Through frequency domain analysis, the author examines how this technique affects different aspects of the image. Based on these findings, they recommend including DIP-based watermark removal in future evaluation frameworks to better assess watermark robustness.

**Audience:**

Yes

**Broader Impact Concerns:**

No Broader Impact Concerns

**Claims And Evidence:**

Yes

**Requested Changes:**

Critical:
1. Please add the evaluations I mentioned in point 2 under Weaknesses.

Non-critical:
1. Please summarize the large tables in more concise ones. It would be nice to include more graphical results in their place.
2. Please add a discussion about the assumption of access to a watermark detector for optimization.

**Strengths And Weaknesses:**

Strengths:
1. The paper presents clear and accessible content throughout most sections.
2. The authors provide extensive background information and thoroughly discuss the motivation behind the research.
3. The effectiveness of the method is analyzed in terms of its impact on different frequency bands.

Weaknesses:
1. The method requires access to a watermark detector for hyperparameter optimization.
2. Several important baselines are omitted from the evaluation, including:
a. A recent regeneration attack that has demonstrated exceptional effectiveness against watermarks [1]
b. An advanced watermarking technique that improves upon tree-ring watermarking and is applicable across all image sources [2]
3. The proposed method does not consistently outperform alternatives but rather excels only in specific scenarios.
4. The extensive tables on pages 9-10 would be more appropriately placed in an appendix. A condensed summary table highlighting key findings would better serve readers in the main text.

[1] Zhao, Xuandong, et al. "Invisible image watermarks are provably removable using generative ai." Advances in Neural Information Processing Systems 37 (2024): 8643-8672.
[2] Arabi, Kasra, et al. "Hidden in the Noise: Two-Stage Robust Watermarking for Images." The Thirteenth International Conference on Learning Representations (2025).

---

> ### Author Response · Authors · 2025-05-01
> **Response to review FSQv - Part 1**
>
> > Concerns over missing baselines.
>
> We thank the reviewer for pointing out [1] as a relevant baseline!  In fact, the methods we benchmarked under the names `VAE` and `Diffuser` in our paper correspond to `VAE-chen2020` and `diffusion model` proposed in [1]. We acknowledge a citation error where `VAE` and `Diffuser` method should be pointed to [1] instead of [4] in the experiment setup in section 4. This will be corrected in our revision.
>
> In terms of work [2], we were unaware of this paper by the time of submission. Also, their first version of open-source code was released on March 13, 2025, whereas our manuscript was submitted on February 19, 2025. So, even if we were aware of their paper, their implementation was not available for a comparison at the time of our initial submission.
>
> Nonetheless, we have performed limited-scope benchmark experiments during the rebuttal period: due to the brevity of the rebuttal period, we can only experiment with a subset of the entire evaluation set, consisting of 50 **randomly chosen** images from the whole set; we will continue the experiments so that we can include the final result in our revision. Below is the preview of the results on the 50 images:
>
> 1. `WIND_inpaint` — a post-processing watermark proposed in [2]. We use `WIND_inpaint` to generate 50 watermarked images from the COCO dataset, decode both the clean images and watermarked images, and then collect the detection distances. Based on the threshold-based decision rule in line 8 of their Alg. 2, we compute the true positive rate (TPR, the rate of watermarked images being correctly detected) and false positive rate (FPR, the rate of clean images being incorrectly detected as watermarked ones) based on different thresholds:
> | Metric | Threshold 140 | Threshold 150 | Threshold 160 | Threshold 170 | Threshold 180 | Threshold 190 | Threshold 200 |
> |:------:|:-------------:|:-------------:|:-------------:|:-------------:|:-------------:|:-------------:|:-------------:|
> |   TPR  |      0.0      |      0.0      |      0.2      |      0.68     |      0.96     |      0.96     |      1.0      |
> |   FPR  |      0.0      |      0.0      |      0.14     |      0.64     |      0.9      |      0.96     |      0.98     |
>
> Based on the above result, `WIND_inpaint` cannot achieve a TPR close to 1 with an FPR close to 0 by any threshold. So, `WIND_inpaint` is not a performant watermarking system in practice, and robustness evaluation of this watermark will not be informative. Note that this type of analysis was not performed in [2], and we are actively confirming with the authors about this observation.
>
> 2. `WIND_fast` — a Tree-Ring like in-processing watermark. First, we analyze the TPR/FPR as above, and `WIND_fast` is able to achieve TPR=1 while FPR=0 with our evaluation on 50 samples. Then, we conduct a robustness evaluation following the protocol of our Tab. 4. Below is the result:
>
> | Threshold      |                       | 120                             | 10000                           |
> |----------------|-----------------------|---------------------------------|---------------------------------|
> | TPR/FPR        |                       | 1 / 0                           | 1 / 0                           |
> | gaussian_noise | (PSNR - SSIM - Quatile) | 23.92 - 0.42 - 29.34            | Not able to evade >= 90% images |
> | jpeg           |                       | **33.67 - 0.90 - 10.22**            | Not able to evade >= 50% images |
> | bm3d           |                       | 28.19 - 0.84 - 11.98            | Not able to evade >= 10% images |
> | Diffuser       |                       | Not able to evade >= 90% images | Not able to evade >= 10% images |
> | Diffpure       |                       | **30.64 - 0.84 - 12.42**            | **23.42 - 0.68 - 30.26**            |
> | DIP            |                       | 18.50 - 0.62 - 70.14            | 11.90 - 0.46 - 108.40           |
>
> We can conclude that `WIND_fast` is robust to most of the evasions but is still vulnerable to `Diffpure` among all the evasion methods we have benchmarked in this paper, which is similar to Tree-Ring.
>
> To summarize, based on preliminary evaluation, including the benchmarking results from [2] will not change our conclusion for `post-processing` watermarks, which is the main focus of our paper. However, it helps to reveal the limitation of DIP evasion on `in-processing` watermarks.
>
> We do not expect the full-scale evaluation that is still ongoing to alter the main observations and conclusions.
>
> > The method requires access to a watermark detector for hyperparameter optimization.
>
> Reviewer 13f4 has a similar question. We kindly refer the current reviewer to the response under the title ““Concerns over reliance on a decoder and being less practical for real-world applications”.

---

> ### Author Response · Authors · 2025-05-01
> **Response to review FSQv - Part 2**
>
> > Concerns about the proposed method not consistently outperforming the alternatives
>
> We thank the reviewer for pointing this out! This is part of the message we hope to deliver in the paper — there is no universal best single evasion method, and most existing evasion methods complement each other. This implies that to design robust watermark systems, one has to consider all these possible evasions that attack the systems from complementary angles. The state-of-the-art watermark benchmarking paper [3] conveys similar messages to ours.
>
> > Suggestion on the layout of tables.
>
> Thank you for your suggestion. We will try to figure out a more space-efficient way in the revision.
>
> ------------------------
>
> [1] Zhao, Xuandong, et al. "Invisible image watermarks are provably removable using generative ai." Advances in Neural Information Processing Systems 37 (2024): 8643-8672.
>
> [2] Arabi, Kasra, et al. "Hidden in the Noise: Two-Stage Robust Watermarking for Images." The Thirteenth International Conference on Learning Representations (2025).
>
> [3] An, Bang, et al. "Benchmarking the robustness of image watermarks." ICML, 2024.
>
> [4] Jiang, Zhengyuan, Jinghuai Zhang, and Neil Zhenqiang Gong. "Evading watermark based detection of ai-generated content." Proceedings of the 2023 ACM SIGSAC Conference on Computer and Communications Security. 2023.

---

> ### Comment · Reviewer_FSQv · 2025-05-05
>
> I thank the authors for their rebuttal.
>
> ***"Concerns over missing baselines."***
>
> Thanks for the clarification regarding [1].
>
> Regarding [2], the table first seems inaccurate: [2] goes beyond watermark detection and shows recovery of one out of `N` keys, enabling practical identification of the watermark among non-watermarked content.
>
> That said, I acknowledge the results concerning the in-processing watermark variant and the submission date; I suggest including a corrected version of the first table in the final manuscript
>
> ***"The method requires access to a watermark detector for hyperparameter optimization."***
>
> Thank you, please clarify this in the final version.

---

### Review · Reviewer_afR8 · 2025-04-12

**Summary Of Contributions:**

This paper explores the application of Deep Image Prior (DIP) for the task of invisible watermark removal. By leveraging DIP’s tendency to reconstruct low-frequency image content before high-frequency details, the authors propose a simple black-box attack that requires no access to the watermarking method, training data, or model parameters. The method shows promising results in removing certain types of watermarks and is positioned as a strong baseline for evaluating watermark robustness. A frequency-domain analysis is used to support the effectiveness of the approach.

**Audience:**

Yes

**Broader Impact Concerns:**

The method reveals a potential vulnerability in some watermarking techniques, but the paper falls short in addressing the ethical and practical implications of enabling watermark removal in the public domain. While the stated goal is to motivate more robust watermark design, this point is not sufficiently emphasized or developed in the manuscript.

**Claims And Evidence:**

Yes

**Requested Changes:**

1. Deepen theoretical and failure analysis: Go beyond intuition to provide more systematic insight into when and why the method works or fails.
2. Expand comparative experiments: Include comparisons with other unsupervised denoising or inpainting techniques to contextualize the performance of DIP.
3. Discuss efficiency tradeoffs: Quantify runtime and clarify whether this method is meant for practical attacks or offline benchmarking only.

**Strengths And Weaknesses:**

Strengths:
1. The approach is refreshingly simple, requiring no model training or specialized architecture.
2. The empirical evaluation is broad, covering different watermarking methods and image datasets.
3. The authors provide some insight into the failure cases and offer informal guidelines for improving watermark design.

Weaknesses:
1. Limited novelty: The method is essentially a direct application of DIP in a new context, without any technical innovation or extension to the original framework.
2. Underdeveloped analysis: The frequency analysis, while intuitive, lacks rigor or quantitative support.
3. Practical limitations unaddressed: The method requires iterative optimization and is computationally expensive. Its use in real-world attack settings or benchmarking pipelines is questionable.

---

> ### Author Response · Authors · 2025-05-03
> **Response to reviewer afR8 --- Part 1**
>
> > Concerns over limited novelty.
>
> We appreciate the reviewers’ thoughtful comments regarding the novelty of our work. While it is true that our paper does not contain any further development of the DIP framework, doing so is **not** our goal. As an applied work, our novelty lies in the novel application of the DIP denoising idea to watermark evasion, a new context.
>
> > Request to deepen the theoretical and failure analysis / quantitative support over frequency analysis.
>
> About deepening the overall theoretical analysis: We thank the reviewer for suggesting this! But, as an applied work, our priority here is to perform a solid experimental study of our new DIP-based method for watermark evasion. We will defer the theoretical analysis as future work.
>
> We have tried to strengthen our frequency analysis, which provides the intuition on why our method works, during this rebuttal period. Specifically, we have performed more quantitative analysis of how the watermark changes the spectral contents. For each image, we divided the Fourier spectrum of the watermark pattern  ($I_w - I$) into 10 radial bands, and then calculated the averaged relative Fourier Band Error (FBE) for each band. The table below presents the average relative FBE induced by different watermark methods, i.e., a quantitative version of Fig 6 and related discussion in our current paper.
>
> | Method     | Band 1 | Band 2 | Band 3 | Band 4 | Band 5 | Band 6 | Band 7 | Band 8 | Band 9 | Band 10 |
> |------------|--------|--------|--------|--------|--------|--------|--------|--------|--------|---------|
> | dwtDctSvd  | 0.025  | 0.044  | 0.059  | 0.074  | 0.091  | 0.103  | 0.109  | 0.109  | 0.110  | 0.371   |
> | rivaGAN    | 0.016  | 0.033  | 0.061  | 0.097  | 0.132  | 0.163  | 0.196  | 0.214  | 0.237  | 0.240   |
> | SSL        | 0.011  | 0.036  | 0.070  | 0.119  | 0.174  | 0.222  | 0.289  | 0.331  | 0.377  | 0.765   |
> | **Trustmark**  | 0.017  | 0.075  | **0.177**  | **0.187**  | **0.189**  | 0.175  | 0.148  | 0.113  | 0.094  | 0.109   |
> | **Rosteals**   | 0.085  | 0.282  | **0.386**  | **0.315**  | **0.411**  | 0.436  | 0.475  | 0.556  | 0.620  | 0.307   |
> | **StegaStamp** | 0.166  | 0.350  | **0.432**  | **0.443**  | **0.492**  | 0.518  | 0.503  | 0.481  | 0.434  | 1.768   |
>
> Combining the result of Table 2 in our current paper and the above table, we can conclude that the watermarks what our DIP-based method struggles to evade, i.e., Trustmark, Rosteals, and StegaStamp, all have considerably large relative FBE on the lower mid-frequency band (Band 3-5).  Especially, comparing SSL (whose FBE has larger magnitudes on Band 6-10) and Trustmark (whose FBE has larger magnitudes on Band 3-5), it is even clearer that leveraging lower frequency bands is the key to counter DIP evasions.
>
> > Concerns over iterative optimization,computation cost; discuss the efficiency tradeoffs.
>
> We kindly refer the current reviewer to our response to reviewer 13f4 on “Concerns over computation efficiency” for a similar concern.

---

> ### Author Response · Authors · 2025-05-03
> **Response to reviewer afR8 --- Part 2**
>
> > Expand comparative experiments: Include comparisons with other unsupervised denoising or inpainting techniques to contextualize the performance of DIP.
>
> First, we want to reiterate that the goal of this paper is not to develop the DIP method further, but to apply it in the novel context of watermark evasion. So, it is reasonable to compare ours with other watermark evasion methods, but not with other DIP applications.
>
> Regarding comparing to inpainting techniques, if the reviewer’s point is about inpainting-based removal of visible watermarks, please refer to our response to reviewer 13f4 on “Concerns of novelty; and questions related to inpainting techniques” for why `inpainting` techniques do not apply to the problem of removing invisible watermarks.
>
> In terms of comparison to `other unsupervised denoising techniques`, we have included the most popular traditional method `bm3d` in our benchmarks, which is a non-data-driven approach. If we take a broader definition of `unsupervised denoising`, which includes “training-based denoising techniques that do not have access to (clean, noisy) image pairs”, our benchmarked method `Diffpure` from [1] also falls into the category. We include these methods, as both have been applied to removing invisible watermarks in previous work. We think our benchmark selection is representative to show that: (1) DIP relies on a completely different principle to remove invisible watermarks from previous approaches; (2) DIP outperforms existing watermark evasion techniques in certain cases, while there is no universal `best` watermark evader; and (3) the inclusion of our DIP-based evasion into the robustness benchmark of image watermarking systems is beneficial due to (1) and (2).
>
> Despite the above response, we are open to the reviewer’s suggestion and kindly ask for some more details and clarification, if any. In particular, we would like to know which paper or technique on `unsupervised denoising` the reviewer thinks is particularly interesting to be included in the benchmark.
>
> > Lack of addressing ethical concerns.
>
> We thank the reviewer for pointing out this aspect that we have overlooked. We will make sure to add one section to discuss the ethical implications of our work.
>
> ---------
>
> [1] Nie, Weili, et al. "Diffusion models for adversarial purification." arXiv preprint arXiv:2205.07460 (2022).

---

### Review · Reviewer_13f4 · 2025-04-16

**Summary Of Contributions:**

This paper investigates the use of Deep Image Prior (DIP) — a technique that leverages the inherent bias of convolutional neural networks (CNNs) for image denoising—as a method for removing watermarks from images. Through extensive experiments, they demonstrate that DIP is also effective at erasing watermarks, particularly those embedded in the high-frequency components of an image. The underlying mechanism is similar to denoising: due to the architectural bias of CNNs, the network tends to reconstruct low-frequency structures first. By employing early stopping, we can prevent the network from fully reconstructing the high-frequency details where watermarks typically reside, effectively removing them from the output.

**Audience:**

Yes

**Broader Impact Concerns:**

No concerns on the ethical implications

**Claims And Evidence:**

Yes

**Requested Changes:**

1. Equation 2 seems to have some problems, there is no constraint on equation (1b).
2. What is the baseline method diffuser? The paper lacks citations over this method.

**Strengths And Weaknesses:**

## Strength

1. The paper is well-written and easy to follow. I appreciate the comprehensive review of related work, which systematically summarizes prior approaches to watermark.
2. The experimental evaluation is solid. The paper benchmarks its method against various watermarking techniques and compares results with several baseline methods. It also clearly acknowledges its limitations, particularly with watermarks embedded in the high-frequency domain—consistent with the theoretical basis of DIP. Although the proposed method does not achieve state-of-the-art performance across all datasets, it still provides valuable insights and contributions to the field.

## Weakness

1. The technical novelty of the paper is limited, especially considering the [prior work](https://github.com/braindotai/Watermark-Removal-Pytorch) cited by the authors. The methodology appears to be largely the same as in the earlier work, which also employed DIP without requiring watermark patterns or location priors. The main distinction seems to be that this paper focuses on invisible watermarks, where DIP performs better—unlike in the prior work that targeted visible watermarks, which were more challenging to remove effectively.

2. The method’s reliance on a decoder makes it less practical for real-world applications, especially since, to my knowledge, none of the baseline methods share this requirement. This constraint significantly limits the usability and generalizability of the approach.

3. This paper lacks a direct computation efficiency comparison of this method and baseline, as DIP requires optimization of thousands of iterations per image.

---

> ### Author Response · Authors · 2025-05-01
> **Response to reviewer 13f4 --- Part 1**
>
> > Concerns of novelty; and questions related to inpainting techniques.
>
> We would argue that the novelty of our work lies in the innovative application of existing techniques to a new problem. To justify why our problem is a new one, we would like to point out a misconception here respectfully: `...the prior work [1] employed DIP without requiring watermark patterns or location priors...` is not true, as [1] removes visible watermarks by DIP-based inpainting—which requires the inpainting mask, i.e., the location priors of the watermark. For our removal of invisible watermarks, we phrase it as an image regression problem that requires **neither the watermark pattern nor location**. So, [1] and our work **target two different problems**: the former on the removal of visible watermarks given the watermarked image and the watermark location, the latter on the removal of invisible watermarks given only the watermarked image—it is arguable the former is more challenging than the latter given that it requires more knowledge, with **two different DIP-based methods**: the former using DIP-based inpainting, and the latter using DIP-based image regression.
>
> > Concerns over reliance on a decoder and being less practical for real-world applications
>
> This seems to be a misconception. When the goal is to bypass the detection of the watermark decoder (i.e., evasion) while obtaining images of the highest quality, all methods considered in our paper, including our method, typically need to query the decoder (in black-box manner) multiple times to finetune their hyperparameters,  e.g., the number of diffusion steps for DiffPure and Diffuser, the compression factor for JPEG, etc. We follow this standard practice to perform fair comparisons with other evasion methods. If the goal is to perform single-shot evasion without worrying about the tradeoff between evasion success and image quality and optimization of the related hyperparameters, then none of these methods need to rely on a decoder.
>
> > Concerns over computation efficiency
>
> With respect to runtime efficiency, we appreciate the reviewer’s concern and will incorporate relevant discussion in the updated version. In general, DIP can operate on lightweight convolutional networks and does not demand excessive computational resources. Below is a snapshot of runtime comparisons performed on a local desktop equipped with an Intel Core i7-12700K processor and an NVIDIA RTX 3080 GPU for processing a single image:
>
> | Method                            | Configuration                          | Time (seconds) |
> |-----------------------------------|----------------------------------------|----------------|
> | Diffuser                          | 10 different diffusion steps           | ~3.85          |
> | BM3D                              | 5 different standard deviation parameters | ~18.2       |
> | DIP [3]                           | 500 iterations                         | ~26.6          |
> | DiffPure                          | 10 different diffusion steps           | ~290.75        |
>
> These results indicate that DIP is on par with Diffuser and BM3D in terms of efficiency and is significantly more computationally efficient than DiffPure. Additionally, we note that there is existing research on accelerating DIP (e.g., [4]), which, although not covered in our current paper, offers promising avenues for enhancing the computational efficiency of DIP.
>
> > Question on the correctness of eq. 2.
>
> Your observation about eq. 2. not including the constraint of eq. 1b  is right. However, this is not a mistake, but a faithful review of all existing learning-based steganography methods — eq. (1b) is a necessary component in the problem formulation, while eq. 2. is the practical implementation of most existing methods, which do not enforce this constraint during training time.

---

> ### Author Response · Authors · 2025-05-01
> **Response to reviewer 13f4 --- Part 2**
>
> > What is the baseline method of diffuser?
>
> Baseline method `Diffuser` is `diffusion model` method proposed in [5]. We noticed a citation error where `VAE` and `Diffuser` method should be pointed to [5] instead of [6] in the experiment setup in section 4. We will correct this in our revision.
>
> ---------------------------
>
> [1] https://github.com/braindotai/Watermark-Removal-Pytorch
>
> [2] How to Remove Dall-E Watermark: https://www.youtube.com/watch?v=6EMROCxGCIA
>
> [3] Ulyanov, Dmitry, Andrea Vedaldi, and Victor Lempitsky. "Deep image prior." Proceedings of the IEEE conference on computer vision and pattern recognition. 2018.
>
> [4] Li, Taihui, et al. "Deep random projector: Accelerated deep image prior." Proceedings of the IEEE/CVF Conference on Computer Vision and Pattern Recognition. 2023.
>
> [5] Zhao, Xuandong, et al. "Invisible image watermarks are provably removable using generative ai." Advances in Neural Information Processing Systems 37 (2024): 8643-8672.
>
> [6] Jiang, Zhengyuan, Jinghuai Zhang, and Neil Zhenqiang Gong. "Evading watermark based detection of ai-generated content." Proceedings of the 2023 ACM SIGSAC Conference on Computer and Communications Security. 2023.

---

### Decision · Action_Editor_GaK5 · 2025-06-01

**Recommendation:** Accept with minor revision

**Comment:**

After a short rebuttal stage, all three reviewers leaned towards acceptance. I personally enjoyed reading the paper. The proposed method is simple and effective, and while the approach echoes the idea established in previous work (e.g. [1]), I agree with the authors that (a) the use of Deep Image Priors (DIP) for watermark evasion is sensible, and (b) DIP-based evasion can be used as a fruitful baseline for future work in this direction. I recommend acceptance and request the authors to update their manuscript to reflect the discussions with the reviewers while preparing the final version.

**Audience:**

Yes, the topic (removal of watermarks using deep image priors) is within scope of TMLR, and the paper presents several interesting findings.

**Claims And Evidence:**

The paper is well written and the authors present compelling evidence supporting their proposed DIP-based method for watermark evasion.